# AZ-whiteness test: a test for signal uncorrelation on spatio-temporal graphs

**Daniele Zambon**
The Swiss AI Lab IDSIA
Università della Svizzera italiana
Lugano, Switzerland
`daniele.zambon@usi.ch`

**Cesare Alippi**
The Swiss AI Lab IDSIA
Università della Svizzera italiana
Politecnico di Milano
Lugano, Switzerland
`cesare.alippi@usi.ch`

## Abstract

We present the first whiteness hypothesis test for graphs, i.e., a whiteness test for multivariate time series associated with the nodes of a dynamic graph; as such, the test represents an important model assessment tool for graph deep learning, e.g., in forecasting setups. The statistical test aims at detecting existing serial dependencies among close-in-time observations, as well as spatial dependencies among neighboring observations given the underlying graph. The proposed AZ-test can be intended as a spatio-temporal extension of traditional tests designed for system identification to graph signals. The AZ-test is versatile, allowing the underlying graph to be dynamic, changing in topology and set of nodes over time, and weighted, thus accounting for connections of different strength, as it is the case in many application scenarios like sensor and transportation networks. The asymptotic distribution of the designed test can be derived under the null hypothesis without assuming identically distributed data. We show the effectiveness of the test on both synthetic and real-world problems, and illustrate how it can be employed to assess the quality of spatio-temporal forecasting models by analyzing the prediction residuals appended to the graph stream.

## 1 Introduction

In the past decade, machine learning methods based on graph-structured data have largely contributed to the field of multivariate time-series analysis with major achievements, *e.g.*, in forecasting and missing values imputation [7, 25]. In this paper, we focus on spatio-temporal time series $\mathbf{x}_v[t], t = 1, 2, \ldots$, positioned[1] at the vertices $v \in V$ of a graph; the functional/structural dependency existing among nodes — say, over space — is modeled by a set $E \in V \times V$ of relations. We denote with $G = (V, E, \mathbf{X})$ the resulting (directed or undirected) *graph* whose nodes $v$ are associated with stochastic time series $\mathbf{x}_v[t] \in \mathbf{X}$, and name $\mathbf{X} = \{\mathbf{x}_v[t] \mid \forall v, t\}$ as a *graph signal*, to be intuitively intended as a set of measurements appended to the graph nodes. The framework can be naturally extended to deal with graphs seen as a realization of a random variable. Graph $G$ can be static, meaning that vertex and edge sets are constant over time, or dynamic, hence modeling frameworks where topology and number of nodes can change. Dynamic graphs appear frequently in cyber-physical systems where sensors can be added or removed, or node data are missing either pointwise or for lapses of time, *e.g.*, following communication or sensor readout faults. Another example is provided by social networks where users expand their set of friends or change preferences and interests over time. Indeed, we account for graphs with weighted edges encoding, for instance,

---

[1]Known as time-vertex signals too, *e.g.*, see [21].

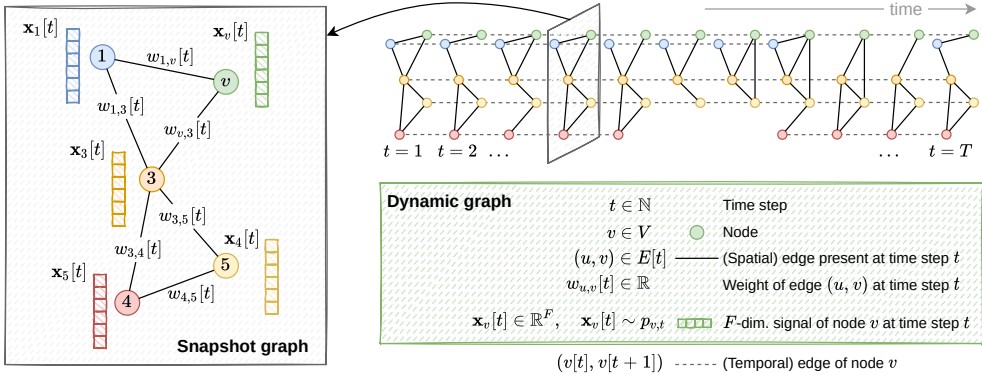

Figure 1: A dynamic weighted graph and associated graph signal. The dynamic graph is defined over a set $V$ of nodes; without requesting all nodes to be always available over time. The topology of the graph is represented by solid lines. Dashed lines represent temporal edges connecting the same node at consecutive time steps. The graph signal at generic time step $t$ and node $v$ is multivariate. On the left-hand side, a single snapshot of the temporal graph and graph signal are extracted.

the capacity or the strength of the link. Figure 1 provides a visualization of a weighted dynamic graph $G$ with associated graph signal $\mathbf{X}$.

In the presented setting we address whether the graph $G$ is *white* or not. We define a white graph as a graph $G = (V, E, \mathbf{X})$ where for all distinct $\mathbf{x}_u[t_i], \mathbf{x}_v[t_j] \in \mathbf{X}$ we have that $\mathbb{E}\left[\mathbf{x}_u[t_i]\right] = \mathbf{0}_F$ is the $F$-dimensional zero vector (with $F$ dimension of the node signals) and $\mathbb{E}\left[\mathbf{x}_u[t_i]\mathbf{x}_v[t_j]^\top\right] = \mathbf{0}_{F \times F}$ is the $F \times F$ zero matrix, in accordance with traditional white signals;[2] as we will see, the graph structure is fundamental asset in designing a whiteness test. In turn, identifying white graphs entails us, for instance, to assess the optimality of a predictive model $f_\theta$ trained to solve a forecasting task associated with stochastic graph signal $\mathbf{X}$. In this prediction application, we consider a residual graph with graph signal $\mathbf{R}$ composed of residuals $\mathbf{r}_v[t] = \hat{\mathbf{x}}_v[t] - \mathbf{x}_v[t]$ evaluated over the estimated values $\hat{\mathbf{x}}_v[t]$ given by model $f_\theta$ and measured ones $\mathbf{x}_v[t]$. Indeed, if the residual graph is not white, there is further information in data not exploited by predictive model $f_\theta$ and the inference problem shows margins for improvement. While assessing whether $\mathbf{X}$ (or $\mathbf{R}$) is zero-mean can be carried out with traditional centrality tests for generic data samples [24], here we focus on the rather unexplored problem of detecting data dependencies within graph signals.

In this manuscript, we propose a novel whiteness test to decide whether a graph $G$ can be considered white or not, *i.e.*, there is prominent evidence of serial (time) and/or spatial correlation among data observations conditioned to the graph structure. In general, testing the assumption of independent signals is an ill-posed problem as it requires studying unknown distributions $p_{v,t}$ (of the stochastic process associated with $\mathbf{X}$) whose number is proportional to the cardinality of nodes and time steps, and for each of which we have a single observation only. We solve this ill-posed problem by integrating into the whiteness assessment method the relational inductive bias associated with the graph topology and the temporal coherence. The AZ-test we propose extends traditional methods for serial correlation [14, 18, 19] to deal with the general case of spatio-temporal correlation and, with this alphabetic encompassing name, we emphasize its widespread applicability. To the best of our knowledge, this is the first paper proposing a general whiteness test. The suggested test statistic is based on counting the positive and negative signs of the product between spatially and temporally adjacent observations, whose disproportion indicates either direct or inverse correlation between a variables pair; indeed, authors can consider different statistics.

The contributions of the paper can be summarized as

- We propose the first statistical test to verify the whiteness hypothesis for a, possibly weighted and dynamic, graph $G$ [Sections 3 and 4].
- We derive the limit distribution of the test statistics under the null hypothesis [Theorem 1].

---

[2]A scalar stationary time series is said to be white if it is zero-mean and the autocorrelation function $\kappa(\tau) = \mathbb{E}[\mathbf{x}[t]\mathbf{x}[t-\tau]] = 0$ for all $\tau \neq 0$ which, in turn, holds if and only if the power density spectrum $S(\omega)$ is constant [20]; the term "white" is in analogy the white light containing all frequencies in the visible spectrum.

Table 1: Configurations in which the proposed whiteness test is applicable.

| Topology | Temp. dimension | Edge weights | Node signals | Correlation |
|----------|-----------------|--------------|--------------|-------------|
| Static | $T = 1$ | Absent | Scalar ($F = 1$) | 1-hop/lag |
| Dynamic | $T > 1$ | Present | Multivariate ($F > 1$) | $K$-hop/lag ($K > 1$) |

- We present a procedure based on the proposed test to assess whether a given forecasting model can be considered optimal with respect to the given data or not [Section 5.2].

The proposed whiteness test is computationally efficient for sparse graphs as the number of operations scales linearly with the number of edges and time steps. Moreover, the test is very general and allows us for incorporating very relevant application contexts and operational scenarios, *e.g.*, those listed in Table 1. In particular, the test is applicable when $\mathbf{x}_v[t]$ is uni- or multivariate, when $G$ is static or dynamic, when the edges of $G$ are weighted, and when the graph signal is composed of a set of time series or it is static, namely, $\mathbf{X} = \{\mathbf{x}_v \in \mathbb{R}^F \mid v \in V\}$, and there is no temporal dimension involved.

The remainder of the paper is structured as follows. Section 2 reviews related work. Section 3 presents the test in the simplified case without the temporal dimension. Section 4 shows how the test designed in Section 3 can be applied to spatio-temporal signals on dynamic graphs. Section 5 reports empirical evidence of the statistical power of the test and shows how to assess the optimality of forecasting models by applying the test to prediction residuals. Finally, Section 6 draws some conclusions and provides pointers to future research. Supplemental material accompanies the manuscript with proofs, extensions of the proposed test, and further experimental details.

## 2   Related work

Among the most renowned whiteness tests, there are the Durbin-Watson test [9] and the Ljung-Box test [19]. Both have been introduced to test serial dependency in a univariate time series. More recently, Drouiche [8] proposed a test that operates on the spectral density of the signal. Several whiteness tests have been proposed for multivariate data as well, *e.g.*, see [2, 5, 14, 16, 18].

To the best of our knowledge, no work has presented a whiteness test for spatio-temporal signals, and what proposed here is pioneering in this direction. A few tests consider a graph structure to represent relationships among the given observations [4, 11, 22], but none of them is a whiteness test. The AZ-test we propose shares the fundamental idea of Geary's test [12] of counting sign changes between consecutive observations in a univariate time series, but introduces several advancements that make it a substantial and original contribution. Apart from the fact we are considering graphs, which is a major contribution per se, the most important result is the applicability of the test to spatio-temporal signals defined over graphs. Secondly, the designed test is general enough to operate on weighted and dynamic graphs with multivariate node signals.

## 3   Whiteness test for static graphs

For the sake of clarity, we first present the test in a simplified — yet relevant — setting characterized by a static weighted graph and a static signal with a single observation (scalar or vector) associated with each node; an example is given in Figure 2. The most general case where time is involved and the graph is dynamic, as in the scenario depicted in Figure 1, is provided in next Section 4 as an extension of what we present here by considering a suitably constructed (multiplex) graph involving temporal edges alongside spatial edges.

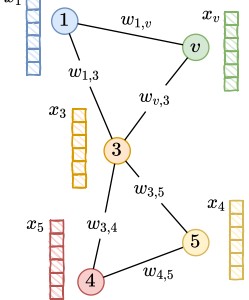

Figure 2: A static graph $G$ with time-independent multivariate node signals.

Consider a weighted graph $G = (V, E, \mathbf{W}, \mathbf{X})$ defined over node set $V$, edge set $E \subseteq V \times V$ without self-loops,[3] and scalar weights

$$\mathbf{W} = \{w_{u,v} \in \mathbb{R}_+ \mid (u, v) \in E\}.$$

---

[3] As we comment below, and in Supplemental Material, we avoid self-loops to simplify the notation only.

The graph topology and weights define the spatial structure underlying a graph signal

$$\mathbf{X} = \{\mathbf{x}_v \in \mathbb{R}^F \mid v \in V\}$$

defined over the nodes of $G$. We assume that the edge weights are positive real numbers that encode the strength/capacity of the links; without loss of generality, we consider absent edges characterized by a weight equal to zero. Node signals $\mathbf{x}_v \in \mathbb{R}^F$ can be scalars ($F = 1$) or vectors ($F > 1$), but they are static, meaning that no temporal information is associated with them, as shown in Figure 2. Essentially, we consider a single snapshot of the dynamic graph depicted in Figure 1.

**The test**   The ultimate goal is to test if graph signal $\mathbf{X}$ is *white* or it displays dependencies among nodes. The statistical hypotheses of the test are

$$\begin{cases} H_0 : & \mathbf{x}_u, \mathbf{x}_v \text{ are uncorrelated for all } u \neq v \in V, \\ H_1 : & \mathbf{x}_u, \mathbf{x}_v \text{ are correlated for some } u \neq v \in V, \end{cases} \tag{1}$$

and the proposed AZ-whiteness test is

$$\text{If } |C(G)| > \gamma \implies \text{ Reject the null hypothesis } H_0, \tag{2}$$

and it is based on threshold $\gamma > 0$ and statistic

$$C(G) = \frac{\widetilde{C}(G)}{\sqrt{W}} \quad \sim \mathcal{N}(0,1) \text{ as } |E| \to \infty, \tag{3}$$

(as we prove in Theorem 1), where $|E|$ denotes the cardinality of set $E$ — that is, the number of edges in $G$ — while $C(G)$ is defined over quantities

$$\widetilde{C}(G) = \sum_{(u,v) \in E} w_{u,v} \, \mathrm{sgn}(\mathbf{x}_u^\top \mathbf{x}_v), \tag{4}$$

$$W = \sum_{(u,v) \in E \setminus E_\leftrightarrow} w_{u,v}^2 + \sum_{(u,v) \in E_\leftrightarrow} (w_{u,v} + w_{v,v})^2 \tag{5}$$

with $\mathrm{sgn}(x) = 1$ if $x > 0$, $\mathrm{sgn}(x) = -1$ if $x < 0$, and $\mathrm{sgn}(x) = 0$ if $x = 0$, and where set $E_\leftrightarrow$ accounts for multiple edges linking the same pair of nodes, that is $E_\leftrightarrow = \emptyset$ for undirected graphs, otherwise $E_\leftrightarrow = \{(u,v) \in E \mid u < v, \ (v,u) \in E\}$.

The intuition behind (2) to test null hypothesis $H_0$ against $H_1$ is that, when the node signals in $\mathbf{X}$ are mutually independent and centered around zero, then random variables $\mathrm{sgn}(\mathbf{x}_u^\top \mathbf{x}_v)$ for all $u \neq v \in V$ are centered in zero too as proven in Lemma 1 (supplemental material). In particular, in the case of scalar signals, $\mathrm{sgn}(\mathbf{x}_u \mathbf{x}_v) = \mathrm{sgn}(\mathbf{x}_u)\mathrm{sgn}(\mathbf{x}_v)$ and we observe that large values of $C(G) \gg 0$ indicate the presence of few sign changes between observations of adjacent nodes, which in turn suggests correlation among variables. Similarly, negative correlation is revealed by $C(G) \ll 0$. Therefore, $|C(G)| \gg 0$ is symptomatic of a dependent data set up, and justifies our test (2).

Theorem 1 supports the soundness AZ-whiteness test (2) and provides a distribution-free criterion to select threshold $\gamma$ granting a user-defined significance level $\alpha = \mathbb{P}(\text{Reject } H_0 | H_0)$, *e.g.*, $\alpha = 0.05$.

**Theorem 1.** *Consider a weighted graph $G = (V, E, \mathbf{W}, \mathbf{X})$ without self-loops and stochastic graph signal $\mathbf{X} = \{\mathbf{x}_v \in \mathbb{R}^F \mid v \in V\}$ on it, with $\mathbf{x}_v \neq 0$ almost surely. Under assumptions*

    **(A1)** *All $\{\mathbf{x}_v \mid v \in V\}$ are mutually independent (hypothesis $H_0$ in (1)),*

    **(A2)** $\mathbb{E}_{\mathbf{x}_v}\left[\mathrm{sgn}(\bar{\mathbf{x}}^\top \mathbf{x}_v)\right] = 0$ *for all $\bar{\mathbf{x}} \in \mathbb{R}^F \setminus \{\mathbf{0}\}$ and $v \in V$,*

    **(A3)** $w_{u,v} \in (0, w_+]$ *for all $(u,v) \in E$, and $W \to \infty$ as $|E| \to \infty$,*

*the distribution of $C(G)$ in (3) converges weakly to a standard Gaussian distribution $\mathcal{N}(0,1)$ as the number $|E|$ of edges goes to infinity.*

In light of above Theorem 1, threshold $\gamma$ is selected to be the quantile $1 - \alpha/2$ of the standard Gaussian distribution so that $\mathbb{P}(|C(G)| > \gamma \mid H_0) = \alpha$, thus ensuring to meet the user-defined significance level $\alpha$.

Before sketching the proof, we comment that, although Assumption (**A2**) is not always compatible with the null-mean hypothesis for $\mathbf{X}$ to be white noise, we need (**A2**) to test data correlation only. Moreover, in the scalar case with $F = 1$, (**A2**) reduces to ask that the median $m$ of all $\mathbf{x}_v$ is zero and, when $m \neq 0$, we can safely[4] and equivalently run the test on $\mathbf{X}' = \{\mathbf{x}_v - m \mid v \in V\}$. Notably, (**A2**) allows node signals to have different distributions. Assumption (**A3**) is technical and takes part in the limit case of $|E| \to \infty$; intuitively, it ensures that all edges bring a tangible contribution to the final statistics $C(G)$. The same theorem can be proven under milder assumptions. Finally, the assumptions of no self-loops and $\mathbb{P}(\mathbf{x}_v = 0) = 0$ are made to simplify the notation only. More comments on the assumptions of Theorem 1 can be found in Section B of the supplemental material.

*Sketch of the proof.* The proof of Theorem 1 is based on the fact that, under (**A1**) and (**A2**), random variables $\mathrm{sgn}(\mathbf{x}_v^\top \mathbf{x}_u)$, for all $(u, v) \in E$, are mutually independent, including edges that share one of the two ending nodes. It follows that statistic $\widetilde{C}(G)$ in (4) is a weighted sum of independent Bernoulli random variables. Finally, we prove that, with bounded weights, the Lindeberg condition (Equation 13, in the supplemental material) holds for $\widetilde{C}(G)$ under (**A3**), and conclude by applying the central limit theorem [1] for independent, but not equally distributed, terms of $\widetilde{C}(G)$. The detailed proof is given in Section A of the supplemental material. $\qquad\square$

Despite the apparently little amount of information required to perform the test (only the sign of the scalar products $\mathbf{x}_v^\top \mathbf{x}_u$ of all edges $(u, v) \in E$), Geary [12] has shown that for scalar time-series (interpretable as a path graph) the test performance was aligned with that of more sophisticated tests. Empirical evidence of the effectiveness of the AZ-whiteness test is given in Section 5.

# 4   Whiteness test for spatio-temporal graphs

In this section, we show how to apply the test designed in Section 3 to a generic spatio-temporal signal associated with a possibly dynamic and weighted graph.

Consider a time frame $[1, T]$ and a discrete-time dynamic graph

$$\vec{G} = \left(\vec{V}, \vec{E}, \vec{\mathbf{W}}, \vec{\mathbf{X}}\right) = \left(\{V_t\}_{t=1}^T, \{E_t\}_{t=1}^T, \{\mathbf{W}_t\}_{t=1}^T, \{\mathbf{X}_t\}_{t=1}^T\right), \tag{6}$$

where the arrow "$\to$" on top of the symbols recalls their temporal nature. For each time step $t$, $(V_t, E_t, \mathbf{W}_t, \mathbf{X}_t)$ is a static graph like the one in Figure 2; when we consider time the reference figure is that of Figure 1. Since we are dealing with node-level time series, we have a correspondence between the nodes of $\vec{G}$ at different time steps and, in general, $V_{t_1} \cap V_{t_2} \neq \emptyset$ if $t_1 \neq t_2$. Note that the possibility of having a variable node set is important to model scenarios such as those related to the integration of new intersections in a street map, removal of a faulty device in a real cyber-physical system, or missing data from smart meters in a power grid. As edge weights can be dynamic too, denote with $w_{u,v}[t] \in \mathbf{W}_t$ the weight associated with edge $(u, v)$ present in the edge set $E_t$ at time $t$. In this new setting, each node $v \in \cup_{t=1}^T V_t$ is associated with a stochastic time series $\mathbf{x}_v[t]$ available for all time steps $t = 1, 2, \ldots, T$, if $v \in V_t$ for all $t$, or only for a subset of $\{1, 2, \ldots, T\}$. This setting is depicted in Figure 1.

Indeed, we can model a spatio-temporal signal $\vec{\mathbf{X}}$ associated with a static graph $(V, E, \mathbf{W})$ simply by assuming in (6) that $V_t = V$, $E_t = E$, and $\mathbf{W}_t = \mathbf{W}$, for all $t$.

**The test**   The test designed in the previous section is able to identify spatial dependencies existing among the different nodes; here, we extend it to deal also with temporal dependencies that might exist among observations at different time steps. The null and alternative hypotheses in (1) are extended to incorporate the temporal dimension as

$$\begin{cases} H_0 : & \text{All pairs } \mathbf{x}_u[t_i], \mathbf{x}_v[t_j] \in \vec{\mathbf{X}}, \text{ with } (u, t_i) \neq (v, t_j), \text{ are uncorrelated;} \\ H_1 : & \text{At least a pair } \mathbf{x}_u[t_i], \mathbf{x}_v[t_j] \in \vec{\mathbf{X}}, \text{ with } (u, t_i) \neq (v, t_j), \text{ is correlated.} \end{cases} \tag{7}$$

---

[4]Testing the correlation in $\mathbf{X}$ or $\mathbf{X}'$ is equivalent, since $\mathbb{E}[(\mathbf{x}_u - \mathbb{E}[\mathbf{x}_u])(\mathbf{x}_v - \mathbb{E}[\mathbf{x}_v])] = \mathbb{E}[(\mathbf{x}_u - m - \mathbb{E}[\mathbf{x}_u - m])(\mathbf{x}_v - m - \mathbb{E}[\mathbf{x}_v - m])]$.

The proposed whiteness test for spatio-temporal graphs is

$$\text{If } |C(G^*)| > \gamma \implies \text{Reject null hypothesis } H_0, \tag{8}$$

where $C(G^*)$ is statistic $C(G)$ in (3), but applied to a different graph $G^*$ defined below: the sums in (4) and (5) are now over all edges of $G^*$, instead of $G$. Graph $G^*$ is a conveniently chosen (static) representation of given dynamic graph $\vec{G}$ so that the theory from Section 3 applies to $C(G^*)$ too (*e.g.*, to choose $\gamma$ according to Theorem 1) to test the uncorrelation of graph signal $\vec{\mathbf{X}}$ on dynamic graph $\vec{G}$; accordingly, test (8) relies on the same assumption set of test (2) but, this time, related to graph $G^*$.

A representation of graph $G^*$ is given in Figure 1, where the nodes of $\vec{G}$ are replicated for all their occurrences across time while preserving their spatial connectivity (solid lines), and temporal edges (dashed lines) are added by connecting corresponding nodes at subsequent time steps. In the remainder of the section, we detail the construction of graph $G^* = (V^*, E^*, \mathbf{W}^*, \mathbf{X}^*)$ from given graph (6).

**Nodes of $G^*$**    Define node set $V^*$ as the disjoint union of $V_t$, for all $t$, that is,

$$V^* = \{v[t] = (v, t) \mid v \in V_t, t = 1, 2, \ldots, T\}.$$

If node $v$ is not present in node set $V_t$, for some $t$, then $(v, t) \notin V^*$. Therefore, the number of nodes $|V^*|$ is $\sum_t |V_t|$; for a static graph $G$, $|V^*| = |V| \cdot T$.

**Edges of $G^*$**    Define edge set $E^*$ as a collection of spatial and temporal edges. The set of spatial edges is the disjoint union of sets $E_t$, for all $t$, that is

$$E_{\mathrm{sp}} = \{(u[t], v[t]) \mid (u, v) \in E_t, \ t = 1, 2, \ldots, T\}.$$

Temporal edges, instead, connect consecutive occurrences of the same nodes:

$$E_{\mathrm{tm}} = \{(v[t], v[t+1]) \mid v \in V_t \cap V_{t+1}, \ t = 1, 2, \ldots, T-1\}.$$

The edge set $E^*$ is then given by $E^* = E_{\mathrm{tm}} \cup E_{\mathrm{sp}}$. Referring to Figure 1, set $E_{\mathrm{sp}}$ collects all edges represented by solid lines, while edges in $E_{\mathrm{tm}}$ are represented as dashed lines.

**Weights of $G^*$**    Define edge weights in $\mathbf{W}^*$ following the construction of $E^*$. Spatial edges $(u[t], v[t]) \in E_{\mathrm{sp}}$ are associated with weight $w_{u,v}[t] \in \mathbf{W}_t$ corresponding to edge $(u, v) \in E_t$. The weight of the temporal edges can be set arbitrarily. In general, when all edge weights have similar values, the spatial correlation contributes more to the final statistic than the temporal one, because $|E_{\mathrm{sp}}| = T |V| d \approx d |E_{\mathrm{tm}}|$, where $d$ is the average node degree. In Section C of the supplemental material, we show how to assign a weight $w_{\mathrm{tm}}$ to every temporal edge and yield balanced spatial and temporal contributions. In addition, Section C presents a straightforward way with which the user can decide how to trade off the impacts of the spatial and temporal edges.

**Signal $\mathbf{X}^*$**    Arrange graph signal $\vec{\mathbf{X}}$ in (6) over the new node set $V^*$ of $G^*$ as follows

$$\mathbf{X}^* = \left\{\mathbf{x}_{v[t]} = \mathbf{x}_v[t] \in \mathbb{R}^F \mid v[t] \in V^*, \text{ with } \mathbf{x}_v[t] \in \mathbf{X}\right\}.$$

To conclude, we observe that when $T = 1$ the statistic reduces to the one of Section 3 and that by preprocessing $G \mapsto G^*$ we are able to apply the very same statistical test (2) in different scenarios, including that of a dynamic graph, as we did in the current section. In Section D of the supplemental material, we discuss another preprocessing of $G$ that allows for considering $K$-hop and $K$-lag correlations.

## 5    Experiments

We consider two experimental setups. In the first one, we study the ability of the AZ-test to detect data dependency of various amplitudes in a controlled environment. In the second one, we apply the test to a prediction problem both on synthetic and real-world data. The code to reproduce the experiments is available at `https://github.com/dzambon/az-whiteness-test`.

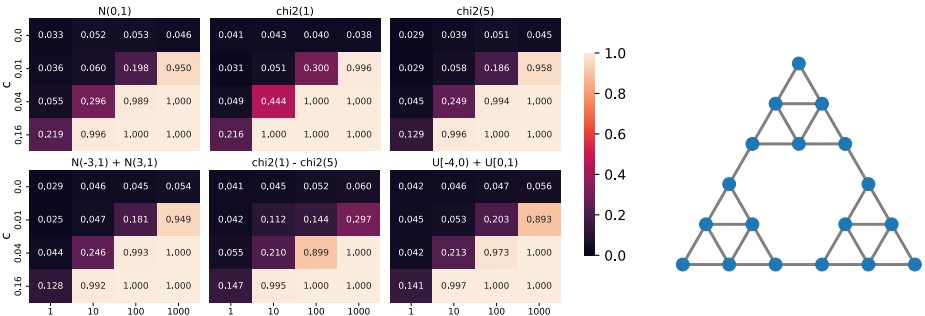

Figure 3: Rate of rejected null hypotheses for significance level $\alpha = 0.05$. Each block corresponds to a different distribution $P$. Every block has different correlation parameters $c$ on the rows and number of time steps $T$ on the columns. The node feature dimension is $F = 1$. The graph underlying the graph signals is drawn on the right-hand side.

## 5.1 Detection of correlated residuals

We consider the undirected unweighted graph $G = (V, E)$ of Figure 3 and denote with $\mathbf{A}$ its adjacency matrix ($\mathbf{A}_{u,v} = 1$ if $(u, v) \in E$ and $\mathbf{A}_{u,v} = 0$, otherwise). In this set of experiments, we generate white-noise and correlated graphs by sampling the components of a matrix $\mathbf{Z} \in \mathbb{R}^{|V| \times T}$ i.i.d. from scalar probability distributions $P$ with null median. Choices of $P$ include unimodal symmetric distributions, as well as asymmetric and bimodal ones; in Figure 3, we consider the standard Gaussian distribution $\mathcal{N}(0, 1)$, the chi-squared distributions[5] $\chi_2(d)$ with $d = 1, 5$ degrees of freedom, and the following mixtures of distributions: $\mathcal{N}(-3, 1) + \mathcal{N}(+3, 1)$, $\chi_2(1) - \chi_2(5)$, and $U[-4, 0] + U[0, 1)$, where $U[a, b]$ uniform distribution in $[a, b]$. In particular, $\vec{\mathbf{X}} = \{\mathbf{x}_v[t] = \mathbf{Z}_{v,t}\}$ is a graph signal of independent observations, meeting the null hypothesis $H_0$ in (7).

Then, we generate correlated signals by propagating an independent signal $\mathbf{Z} \in \mathbb{R}^{|V| \times (T+1)}$ across graph $\vec{G}$ and time as

$$\mathbf{X} = \mathbf{Z}_{t>1} + c_{\mathrm{tm}} \mathbf{Z}_{t \leq T} + c_{\mathrm{sp}} \mathbf{A} \mathbf{Z}_{t>1} - m, \tag{9}$$

where $\mathbf{Z}_{a < t \leq b}$ indicates the reduction of $\mathbf{Z}$ to the columns associated with time steps $t \in (a, b]$. The correlation between observations is controlled by the positive parameters $c_{\mathrm{sp}}$ and $c_{\mathrm{tm}}$. Finally, a scalar offset $m$ is subtracted to have a null-median process $\mathbf{X}$, as requested by the test. We generate graph signals for variable number $T$ of time steps, dimension $F$ of the node features, and correlation parameters $c_{\mathrm{sp}}$ and $c_{\mathrm{tm}}$; from Figure 5 in the supplemental material, it is evident that the higher the values of $c_{\mathrm{sp}}$ and $c_{\mathrm{tm}}$, the more correlated the signals appear on both the temporal and spatial axes.

All experiments in the current section are repeated 1000 times, each time generating a different sequence and applying test (8) with threshold $\gamma$ set to meet a user-defined significance level $\alpha = 0.05$. The considered figure of merit is the rate of rejected null hypotheses, that is, the number of times the test statistic is greater than $\gamma$ over the total number of repetitions of the same experiment.

**Power of the test**   Figure 3 shows the ability of the test to identify deviations from the null hypothesis for different distributions $P$, number of time steps $T$, and values of parameter $c$, with $c_{\mathrm{sp}} = c$, $c_{\mathrm{tm}} = c\, d$ and where $d$ is the average node degree. We observe that (i) the higher the value of $c$, the easier the AZ-whiteness test identifies the correlation, and (ii) the test becomes more powerful as the number $T$ of time steps increases.

**Rate of false positives**   From Figure 3, we can also analyze whether selected test threshold $\gamma$ yields the desired rate of false positives, namely, significance level $\alpha = 0.05$. We see that when $c = 0$ (independent signals) the rate of rejected null hypotheses is around the predefined significance level $\alpha$, regardless of the data distribution $P$ (symmetric/asymmetric, unimodal/bimodal), hence suggesting that $\gamma$ is chosen correctly.

Similar results are displayed in Figure 7 in the supplemental material, where we varied the node feature dimension $F$ too.

---

[5]Distributions $\chi_2(d)$ are shifted to have null median.

## 5.2 Optimality of forecasting models

As a second set of experiments, we train state-of-the-art forecasting models and assess the whiteness of the residual graph signal; when it is not the case, *i.e.*, the residual signal is colored, then predictors have margin to be improved.

### 5.2.1 Experimental setting

We consider three datasets.

**GPVAR** is a synthetic dataset generated from a graph polynomial vector autoregressive system model [15]. The model generates each observation $\mathbf{x}_v[t] \in \mathbb{R}$ at time $t$ and node $v \in V$ from $Q \in \mathbb{N}$ observations in the past and $L$-hop neighboring nodes, with $L \in \mathbb{N}$, as follows:

$$\mathbf{x}[t] = \tanh \left( \sum_{l=0}^{L} \sum_{q=1}^{Q} \Theta_{l,q} \mathbf{S}^l \mathbf{x}[t-q] \right) + \mathbf{z}[t] \qquad (10)$$

where $\mathbf{x}[t] \in \mathbb{R}^N$ concatenates all scalar node signals at time step $t$, $\mathbf{S}$ is a graph shift operator, $\Theta \in \mathbb{R}^{(L+1) \times Q}$ collects the model parameters, and $\mathbf{z}[t] \in \mathbb{R}^N$ is white noise generated form the standard Gaussian distribution [Section 5.1]. Specifically, we consider the undirected and un-weighted graph shown in Figure 4 with the following shift operator $\mathbf{S} = \mathbf{D}^{-1/2}(\mathbf{I} + \mathbf{A})\mathbf{D}^{-1/2}$ where $\mathbf{A}$ is the adjacency matrix (with now self-loops), $\mathbf{D}$ is the diagonal degree matrix, and $\mathbf{I}$ the identity matrix. Model parameters $\Theta$ are set to



Figure 4: Graph of GPVAR dataset.

$[[5,2],[-4,6],[-1,0]]^\top$ and result in $L = Q = 2$. We generate $T = 30000$ time steps.

**PemsBay** is a traffic dataset collected by the California Transportation Agencies Performance Measurement System (PeMS) [17]. It presents $T = 52128$ scalar observations from $N = 325$ sensors in the Bay Area.

**MetrLA** is a dataset containing traffic information from $N = 207$ detectors along the Los Angeles County highway [17]. Observations are collected for 4 months and amount to $T = 34272$ time steps.

The task to solve is a 1-step-ahead forecasting problem where, for every $t$, we want to predict graph signal $\mathbf{x}[t]$ from window $\{\mathbf{x}[t-j] \in \mathbb{R}^N \mid j = 1, \ldots, \tau\}$; $\tau$ is the size of the considered temporal window. In addition, for PemsBay and MetrLA, a positional encoding of the day of observation is added as exogenous variable. On the above forecasting tasks, we trained a Graph WaveNet [25] (GWNet), a Gated Graph Network [23] (GatedGN), and a Diffusion Convolutional RNN [17] (DCRNN). We also consider a baseline model (FCRNN) composed of a 1-layer fully-connected encoder $\mathbf{h}[t] = f(\mathbf{x}[t]; \mathbf{u}[t])$ processing graph signal $\mathbf{x}[t]$ at each time step $t$ and the associated exogenous variables $\mathbf{u}[t]$, and a 2-layer GRU decoder applied to the resulting window of representations $[\mathbf{h}[t-\tau], \ldots, \mathbf{h}[t-1]]$. The size $\tau$ of the temporal window is set to 12 for all experiments. Train, validation and test sets contain 0.7, 0.1, and 0.2 of the original data. The models are trained until convergence with patience of 50 epochs. All models and datasets are available in the TorchSpatiotemporal library [6], except for GPVAR dataset. Further details are available as supplemental material.

### 5.2.2 Results

In Table 2, we report the results for all datasets. In particular, we report the mean absolute deviation (MAE) achieved by the above models on the different datasets and the results of a statistical test on the median of the residuals; MAE and residual median are related in that the minimum of the MAE should produce residuals with null median. The test on the median is implemented as a test on the parameter of Bernoulli random variable $\text{sgn}(\mathbf{x}_v[t])$. Then, we report the results produced by the proposed AZ-whiteness test. In addition to the AZ-test run on the given graph (referred to as "Spatio-temporal" in Table 2), we consider two modified versions to assess the impact of the temporal and spatial components alone, and named "Temporal", where we operate on the temporal edges only by setting to zero the weights of all spatial edges, and "Spatial", where we consider the spatial edges only by setting the weights of all temporal edges to zero; note that the three test statistics are

Table 2: Analysis of observed residuals. The tests of null median report the estimated median while the AZ-tests report the statistic $C(G^*)$; associated $p$-value are subscripted. Bold results highlight results with $p$-value $> 0.01$. Values reported as 0.000 are intended as $< 0.001$.

| Dataset | Model | MAE | Test Median=0 | AZ-test Spatio-temporal | AZ-test Temporal | AZ-test Spatial |
|---|---|---|---|---|---|---|
| GPVAR | Optimal Pred. | 0.319 | **0.001** $_{0.083}$ | **-0.8** $_{0.416}$ | **-0.9** $_{0.355}$ | **-0.2** $_{0.823}$ |
| GPVAR | FCRNN | 0.385 | 0.003 $_{0.010}$ | 5.0 $_{0.000}$ | 8.9 $_{0.000}$ | **-1.8** $_{0.067}$ |
| GPVAR | GWNET | 0.324 | 0.004 $_{0.000}$ | **0.3** $_{0.709}$ | **0.3** $_{0.706}$ | **0.1** $_{0.881}$ |
| GPVAR | GATEDGN | 0.321 | 0.008 $_{0.000}$ | **1.3** $_{0.172}$ | 2.7 $_{0.006}$ | **-0.8** $_{0.414}$ |
| GPVAR | DCRNN | 0.328 | 0.013 $_{0.000}$ | **-0.0** $_{0.955}$ | **-0.6** $_{0.534}$ | **0.5** $_{0.587}$ |
| PemsBay | FCRNN | 2.016 | 0.032 $_{0.000}$ | 1107.4 $_{0.000}$ | 1035.1 $_{0.000}$ | 531.0 $_{0.000}$ |
| PemsBay | GWNET | 0.841 | -0.003 $_{0.000}$ | 422.7 $_{0.000}$ | 7.1 $_{0.000}$ | 590.7 $_{0.000}$ |
| PemsBay | GATEDGN | 0.838 | 0.018 $_{0.000}$ | 454.6 $_{0.000}$ | 25.2 $_{0.000}$ | 617.7 $_{0.000}$ |
| PemsBay | DCRNN | 0.845 | -0.004 $_{0.000}$ | 433.0 $_{0.000}$ | 14.2 $_{0.000}$ | 598.1 $_{0.000}$ |
| MetrLA | FCRNN | 2.842 | -0.016 $_{0.000}$ | 415.3 $_{0.000}$ | 238.5 $_{0.000}$ | 348.8 $_{0.000}$ |
| MetrLA | GWNET | 2.115 | 0.014 $_{0.000}$ | 162.6 $_{0.000}$ | -6.5 $_{0.000}$ | 236.5 $_{0.000}$ |
| MetrLA | GATEDGN | 2.151 | 0.010 $_{0.000}$ | 200.1 $_{0.000}$ | **2.2** $_{0.022}$ | 280.7 $_{0.000}$ |
| MetrLA | DCRNN | 2.141 | -0.018 $_{0.000}$ | 177.4 $_{0.000}$ | 6.7 $_{0.000}$ | 244.1 $_{0.000}$ |

comparable among each other because, as shown in Section C of the supplemental material, they are all asymptotically distributed as a standard Gaussian distribution $\mathcal{N}(0,1)$.

We start by considering the synthetic dataset GPVAR. In this experiment, we know the data generating process (10) and we are able to contrast the considered methods against the optimal predictor, *i.e.*, the graph polynomial VAR filter with the same parameter $\Theta$ that generated the data. From the first row of Table 2, we discover the value of the reference (optimal) MAE and observe that the residual median can be considered null. Then, we note that the outcomes of the AZ-test suggest uncorrelated residuals. Baseline model FCRNN produces a MAE substantially higher than the reference MAE, non-null median of the residual, and correlated residuals. From the Temporal and Spatial variations of the AZ-test, we see that the temporal correlation appears more prominent than the spatial one for FCRNN. In contrast, we can consider the training of GWNet, GatedGN, and DCRNN successful because they produce MAE close to the target value and relatively high $p$-values in the AZ-tests.

Regarding PemsBay and MetrLA datasets, we do not have a reference MAE, as the optimal predictor is unknown. None of DCRNN, GatedGN, and GWNet produces uncorrelated residuals, however, looking at the test statistics, GWNet, GatedGN, and DCRNN display lower temporal correlation than spatial. We point out that the null hypothesis is not rejected as a consequence of Assumption (**A2**) being invalid. We verified it in the supplemental material, where we contrasted the outcome of the analysis conducted on the residuals of FCRNN, GWNet, GatedGN, and DCRNN (Table 2) with that on the same residual, but with the empirical median subtracted (Table 3). For all models and datasets, we obtain almost identical results. We conclude that there is further information left in the residuals that a model can, in principle, learn.

## 6 Conclusions

In this work, we propose the first whiteness test for spatio-temporal time series defined over the nodes of a graph. By exploiting the known structural and functional relations defined by the underlying graph, we are able to identify both temporal and spatial correlations that exist among data observations. We show that the test is asymptotically distribution-free with respect to the number of time steps and graph edges and, therefore, we can apply to data coming from arbitrary distributions and, even, nonidentically distributed observations.

The designed test is general, from which the definitive alphabetic encompassing name, and based on sign changes between observations that are close in time or with respect to the graph connectivity. The test is very versatile, too, as it can exploit edge weights encoding the strength of the link, allows the graph topology to vary over time, and can operate when nodes are inserted or removed from the graph;

these scenarios are frequent when dealing with real-world graph signals, like those coming from sensor networks, *e.g.*, transportation networks, and smart grids. Finally, the test is computationally scalable for sparse graphs with complexity linear in the number of edges and time steps.

We empirically show that the proposed test is indeed capable of identifying dependencies among graph signals; when applied to analyze prediction residuals, the AZ-test can assess whether given forecasting models can be assumed to be optimal or not.

The proposed whiteness test is pioneering in its nature, and allows for future extensions involving stochastic graphs and more sophisticated edge statistics.

### Acknowledgements

This work was supported by the Swiss National Science Foundation project FNS 204061: *HORD GNN: Higher-Order Relations and Dynamics in Graph Neural Networks*.

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
