# OpenReview forum: "AZ-whiteness test: a test for signal uncorrelation on spatio-temporal graphs"
_NeurIPS.cc/2022/Conference — NeurIPS 2022 Accept_

### Official Review · Reviewer_8o7F · 2022-07-09

**Rating:** 6
**Confidence:** 3
**Soundness:** 3 good
**Presentation:** 3 good
**Contribution:** 2 fair

**Summary:**

The paper describes an asymptotic test of whiteness for graph signals that can be used to assess graph signal forecasting models. The main idea is that a whiteness test for graph signals is useful for evaluating forecasting models. Since the model is optimal (in a second order sense) if the residuals are white such that there is no bias and there is no more correlation in the error that the model could account for.  The test is general under the key assumptions including that the signals across space and time can be considered one large graph that is growing in nodes and edges. This generality allows for dynamic graph topology. The test is demonstrated on synthetic and real world datasets.




**Questions:**

Suggestions:

The idea of appending the signal to the graph and defining G as including the signal can lead to subsequent confusion.

Description of whiteness on line 36 is not precise. The quantities involved are vectors or matrices (assuming an outer product) but zero is a scalar. (From the figures row vectors are denoted which means that the second expectation in line 36 is a inner product, but this doesn't make much sense to me. Later on line 110 uses an inner product, so the reader can now be assured that a correlation matrix is intended on line 36.) Furthermore the precise nature of the stochastic time series is not made clear. The stated assumption in the footnote is that the signal is stationary. However, in line 36 it is not clear if the expectations are functions of time or only the time lag like in the footnote. In practice, it is unlikely to have assess to multiple realizations for estimation so assumptions on ergodicity/stationarity should be made, but this point isn't made until around line 52. Also in the footnote it is not clear that $\mathbf{x}$ is multivariate or univariate.

Line 102: Is does null equal zero for the weights? I'm trying to see if this is subtle variation or if a null is somehow distinct from zero.

On line 59, it is not clear what is meant by "; indeed, authors can consider different statistics."

"Supplemental material accompanies the manuscript." It would be useful to detail what this is rather than the blanket statement as provided.

Line 109: It would be better for continuity to use ", where" or equivalent instead of ending the sentence "Theorem 1".

Line 156: I think the G should have an arrow on top in equation 6.


The number of trials used to compute the power is a bit too small. The numbers for the null hypothesis vary. A safer choice of 1000 may enable the reader to better assess whether the false positive rate.

On lines 290 and 291 the comment about p-values being significantly different from zero should be altered to simply say that the null hypothesis cannot be rejected.

Reference list has some problems with format namely book titles should be in title case. [1] appears to be missing punctuation.  The publication type for [12] Geary's test is unclear.

Questions:

In Theorem 1, what is the relationship of the edge cardinality and the node cardinality?
Part of my confusion is that Theorem 1 seems to rely on a distribution over random graphs. For the number of edges to go to infinity, the number of nodes would also have to. But this initially seems to involve a distribution over graph topologies not really a distribution over graph signals. Assumption 3 is important but it described initially as simply being 'technical'. (Line 139 mentions milder assumptions, but it is not clear if that is directly relating to Ass3 or something else.  ) Later on I understand this to be formed from linking nodes through time to have a growing graph. In Section 4, the reader can infer that the growth in edges/nodes required for Ass3 practically corresponds to product graphs that grow through time.

In Table 2 it is not clear why the optimal predictor has a higher MAE than the models.  Additionally the DCRNN model has a significantly non-zero median but shows uncorrelated residuals. Should a user always use both tests to assess modeling performance?  Finally, the description states that the test statistic is not valid since Ass2 is not held. Overall, these mixed results are not very encouraging on the practical usefulness of the method.







**Limitations:**

It is not clear if there is performance thresholds for large number of vertices and a few temporal snapshots.

The paper is perhaps missing baselines for tests of whiteness for signals on regular grids (time series or spatial-temporal forecasting in geospatial settings) where other tests can be readily applied. In these cases, higher resolution (more nodes) can be readily be obtained.

**Strengths And Weaknesses:**

Strengths:
The utility of a robust whiteness measure as the signature of correct modeling of graph signals has practical utility and significance. The paper introduces a general test that is scalable and has an asymptotic characterization. The use of only the sign of the inner product and weights is straightforward and yields a simple asymptotic distribution. The papers coverage of the development and application is thorough. Many more results are mentioned in the supplemental material. The language is easy to read and relations to prior work are stated.

Weaknesses: A key critique is the occasional lack of precise definitions  (especially at the onset) and implicit assumptions that are only revealed after a careful reading. Thus the paper is not clear what is required and how the asymptotic nature is practically realized.

Another critique is that the practical application to real world data and models seems to have mixed results in terms of the model performance (MAE), median of residuals, and the proposed test.

Not until the conclusion do we learn the alphabetic nature of the name. It wasn't apparent to me throughout...

---

> ### Author Response · Authors · 2022-08-02
> **Response to Reviewer 8o7F -- Part 1/3**
>
> Thank you for the careful reading of the paper and the several suggestions that we implemented in the updated manuscript and material. Answers below.
>
>
> ### Definitions and assumptions
>
> > A key critique is the occasional lack of precise definitions (especially at the onset) and implicit assumptions that are only revealed after a careful reading.
>
> We are sorry if you perceive a lack of some assumptions/definitions. We comment that assumptions and main definitions are given (others are considered part of the literature given text constraints). More specifically,
> - Preliminary assumptions (problem setup) related to the test for static graphs (matter of Section 3) are given in Section 3, lines 99-104.
> - Then, Theorem 1, reports the three assumptions requested by the asymptotic distribution of the test statistic to hold.
> - Regarding the test for spatio-temporal graphs (Section 4), the assumptions are exactly those of Section 3, now applied to graph $G^*$.
>
> We clarified this latter point in Paragraph "The test", Section 4. We are happy to further work on the manuscript should you still perceive our answer and intervention as not enough.
>
> > Thus the paper is not clear what is required and how the asymptotic nature is practically realized.
>
> The asymptotic nature of Theorem 1 with respect to the number of edges comes from the central limit theorem employed in the proof (Section A, supplemental material) and requests a sufficient number of edges to have a well-calibrated test threshold $\gamma$;
> the central limit theorem is here applied to a (weighted) sum of edge scores (Eq. 4) upon which statistic in Eq. 3 is built.
> In practice, a sufficient number of edges $|E^*|$ can be obtained by collecting data for a sufficiently large period of time (number $T$ of time steps); in fact, for dynamic graphs $|E^*|=|E_{sp}|+|E_{tm}|\approx T v d + (T-1) v$, where $v=1/T \sum_{t=1}^T |V_t|$ is the average number of nodes in the dynamic graph and $d$ is the average node degree.
>
>
>
> ### Number of trails to assess the statistical power
>
> > The number of trials used to compute the power is a bit too small. The numbers for the null hypothesis vary. A safer choice of 1000 may enable the reader to better assess whether the false positive rate.
>
> Thanks for the suggestion. The results in Figures 3 and 7 now are computed over 1000 repetitions. For convenience, we mention here that a 99% confidence interval for the false positive rate associated with $\alpha=0.05$ and 1000 repetitions is $(0.033, 0.069)$ and we can see that as the number $T$ of time steps increases, the false positive rates are contained in the confidence interval.
>
>
> ### Relationship between edge and node cardinality
>
> > In Theorem 1, what is the relationship of the edge cardinality and the node cardinality? Part of my confusion is that Theorem 1 seems to rely on a distribution over random graphs. For the number of edges to go to infinity, the number of nodes would also have to. But this initially seems to involve a distribution over graph topologies not really a distribution over graph signals.
>
> The test statistic is based on a weighted average of edge scores (see Eq 4), and the core of Theorem 1 is a central limit theorem of that weighted average. Therefore, the asymptotic distribution is intended in the limit of the number of edges (the addends). As the reviewer noticed, for a static graph such an increasing number of edges entails an increase in the number of nodes in $V$, in fact, $|E|\le |V|^2$ (with $|\cdot|$ representing the cardinality of the set); the same can be said for a dynamic graph $\vec G$ by considering the associated static representation $G^*$.
>
>
>
> ### Assumption 3
>
> > Assumption 3 is important but it described initially as simply being 'technical'. (Line 139 mentions milder assumptions, but it is not clear if that is directly relating to Ass3 or something else. ) Later on I understand this to be formed from linking nodes through time to have a growing graph. In Section 4, the reader can infer that the growth in edges/nodes required for Ass3 practically corresponds to product graphs that grow through time.
>
> Assumption 3 can be relaxed as it is only a sufficient condition for the Lindeberg one (Eq 13 and line 473 --now line 480, supplemental material) to hold.
> We refer to Assumption 3 as "technical" because plays a role in the asymptotic regime, while in practice -- i.e., with a finite number of data -- it is enough to have that all weights are positive values.

---

> ### Author Response · Authors · 2022-08-02
> **Response to Reviewer 8o7F  - Part 2/3**
>
>
>
> ### Improvements to the presentation
>
> > Another critique is that the practical application to real world data and models seems to have mixed results in terms of the model performance (MAE), median of residuals, and the proposed test.
>
> We think that a single table reporting all results in the same table is convenient for the reader. We are happy to consider alternatives if the reviewer has better solutions to share with us.
>
> > Not until the conclusion do we learn the alphabetic nature of the name. It wasn't apparent to me throughout…
>
> Thanks for pointing it out to us. In the new paper revision, we mention it already in the introduction.
>
> > The idea of appending the signal to the graph and defining G as including the signal can lead to subsequent confusion.
>
> It is not uncommon to consider attributed graphs, like graphs with node information associated with each of the nodes, instead of considering the graph and the time series as separated entities; for example, see e.g.:
> - Ma et al. 2021. A comprehensive survey on graph anomaly detection with deep learning. IEEE Transactions on Knowledge and Data Engineering.
> - Bacciu et al. 2020. A gentle introduction to deep learning for graphs. Neural networks.
> - Battaglia et al. 2018. Relational inductive biases, deep learning, and graph networks. ArXiv.
>
> > Description of whiteness on line 36 is not precise. The quantities involved are vectors or matrices (assuming an outer product) but zero is a scalar. (From the figures row vectors are denoted which means that the second expectation in line 36 is a inner product, but this doesn't make much sense to me. Later on line 110 uses an inner product, so the reader can now be assured that a correlation matrix is intended on line 36.)
>
> We realize the notation can be improved and we edited the paper accordingly. In particular
> - indeed, in line 36 we are dealing with vectors and matrices -- we removed the abuse of notation;
> - the visual representations in Fig 1 and Fig 2 now show node signals as column vectors in compliance with the rest of the paper.
>
> >  Furthermore the precise nature of the stochastic time series is not made clear. The stated assumption in the footnote is that the signal is stationary. However, in line 36 it is not clear if the expectations are functions of time or only the time lag like in the footnote.
>
> As an example, data may come from a sensor network with associated measurement uncertainty/randomness. Line 36 reads “_for all distinct_ $x_u[t_i], x_v[t_j]\in X$". By this we mean "_for every pair of time steps_ $t_i,t_j$ _and any pair of nodes_ $u,v$ _such that_ $(t_i,u)\ne (t_j,v)$"; clearly, we also assume that both observations $x_u[t_i], x_v[t_j]$ are available (that is, they are not missing values in the time series).
>
>
> > Also in the footnote it is not clear that x is multivariate or univariate.
>
> The beginning of the footnote on page 2 states "_A scalar stationary sequence_", therefore we mean that $\mathbf x$ is univariate, here.
>
> > Line 102: Is does null equal zero for the weights? I'm trying to see if this is subtle variation or if a null is somehow distinct from zero.
>
> We simply mean that the weight is zero. We have now clarified it in the paper.
>
> > On line 59, it is not clear what is meant by "; indeed, authors can consider different statistics."
>
> We mean that the proposed statistic based on the sign function is only one of the possible choices to assess the presence of dependencies in the data.
>
> > - "Supplemental material accompanies the manuscript." It would be useful to detail what this is rather than the blanket statement as provided.
> > - Line 109: It would be better for continuity to use ", where" or equivalent instead of ending the sentence "Theorem 1".
> > - Line 156: I think the G should have an arrow on top in equation 6.
> > - On lines 290 and 291 the comment about p-values being significantly different from zero should be altered to simply say that the null hypothesis cannot be rejected.
> > - Reference list has some problems with format namely book titles should be in title case. [1] appears to be missing punctuation. The publication type for [12] Geary's test is unclear.
>
> Thanks for noticing, we edited accordingly. Regarding the title format of the referenced books, we complied with the [abbrvnat](https://www.bibtex.com/s/bibliography-style-natbib-abbrvnat/) bibliography style.

---

> ### Author Response · Authors · 2022-08-02
> **Response to Reviewer 8o7F  - Part 3/3**
>
> ### Results of Table 2
>
> > In Table 2 it is not clear why the optimal predictor has a higher MAE than the models.
>
> We apology. We discovered that, unfortunately, the first line of results from table 2 and referred to "Optimal Pred." was extracted from the wrong logging file by the script generating the tables; this is also why the same results are reported in Table 3 of the supplemental material. The amended values are available in the new version of the paper. Thank you for having spotted that out!
>
> If the reviewer would like to double-check personally, the correct results can be easily generated by running the submitted code and looking at the log produced by lines 350 and 363 in the script `tsl_experiments.py`. To do so, we suggest to (1) uncomment lines 340-363 in `tsl_experiments.py`, (2) comment out lines 300, 311, and 312 to avoid unnecessary model training, (3) create folder `results` if not already present, (4) run `python tsl_experiments.py --dataset-name gpolyvar`
>
>
> > Additionally the DCRNN model has a significantly non-zero median but shows uncorrelated residuals. Should a user always use both tests to assess modeling performance? Finally, the description states that the test statistic is not valid since Ass2 is not held.
>
> A zero median is not a necessary condition for uncorrelated residuals. However, our test will fail if the median is particularly off; for example, a univariate time series of iid observations distributed according to a Gaussian $N(\mu=10,\sigma=1)$ is uncorrelated (by our iid assumption), but our test would fail because almost all consecutive observations have the same (positive) sign.
>
> In this paper, we check whether the median can be assumed zero or not as the asymptotic distribution is derived under such a condition. We suggest (as we do in the paper) running both tests to check if assumption Ass2 is invalid hence causing the AZ-test to give false results. We refer the reviewer also to Paragraph Assumption (Ass2), Section B, supplemental material, and to the results of Table 3, supplemental material.
>
>
> ### Other comments
>
> > It is not clear if there is performance thresholds for large number of vertices and a few temporal snapshots.
>
> We haven’t experienced any such performance thresholds.
>
>
> > The paper is perhaps missing baselines for tests of whiteness for signals on regular grids (time series or spatial-temporal forecasting in geospatial settings) where other tests can be readily applied. In these cases, higher resolution (more nodes) can be readily be obtained.
>
> The proposed test is the first to be of such general applicability and we provided several empirical results showcasing what can be achieved with our test in setups where all other methods are not applicable. We agree that a comparative analysis in a more traditional problem setting would be insightful, however, we leave it as future work, as we believe that it is out of the scope of the current paper.

---

> > ### Comment · Reviewer_8o7F · 2022-08-07
> > **Response to authors' replies**
> >
> > I appreciate the thorough rebuttal of my review and the other reviewers. I think the thoroughness increases the quality of the paper.

---

### Official Review · Reviewer_LJnR · 2022-07-11

**Rating:** 6
**Confidence:** 3
**Soundness:** 4 excellent
**Presentation:** 3 good
**Contribution:** 3 good

**Summary:**

This paper proposed a whiteness test to examine whether a given graph is "white" or not. The designed test extends classical methods so that it can deal with general case of spatio-temporal correlation. This method applies for uni or multivariate node signals, statistic or dynamic grah, (possibly) weighted edges. The author also show that this test can be used to access the optimality of a given forecating model, which is an interesting and important application.

**Questions:**

1. The distribution of testing statistic is only characterized when the number of edges $|E|\to\infty$. Does it mean this test may not be powerful for small graph? Is there any possible idea to address this issue?
2. In section 4, to perform whiteness test for dynamic and weighted graph, the author argued that we should pre-process it to make it static and then previous approach in Section 3 applies. Would this pre-processing procedure bring increase of type-I/type-II error for AZ-test? Is there any pre-processing procedure and which one is optimal? From my view point, this procedure seems too ad-hoc, and I hope to see some theoretical results behind this procedure.

**Limitations:**

1. Some notations and terminologies are introduced right after they first appeared, e.g., definition of $\text{sng}, E_{\leftrightarrow}, |E|, \rightarrow$. At least the author should create a notation subsection part in Section 1 to introduce all special notations that will be frequently used. If possible, the author may create a preliminary section in main content or supplementary to talk about basics of static/dynamic graph in detail, instead of briefly touching the basics in Section 3/4.
2. Assumptions (Ass1)-(Ass3) are important and repeatedly discussed in later section. The author are suggested to create a special "Assumption" environment to include them instead of putting them in Theorem 1. Also, comments of the assumption can be moved from supplementary to the main content if the paper gets accepted.

**Strengths And Weaknesses:**

**Originality**: This is the first whiteness test that applies for spatio-temporal signals on graph. The design of this test share some idea from Geary' test, but make some additional advancements. Moreover, the designed test is general enough to deal with weighted and dynamic graphs with multivariate node signals.

**Quality/Significance**: From theoretical perspective, the test procedure together with the limiting distribution of testing statistic is derived. I have checked the proof of their theoretical result in detail, and I believe it is clear and correct. From practical perspective, the author examine the ability of AZ-test in various settings and present an interesting application that it can examine the optimality of forecating model. Both parts are excellent contributions.

**Clarity**: The overall presentation is good. However, minor points can be updated to make it more reader-friendly. See some comments in the Limitation section.


**Weakness**:
1. The author pointed out this test share some insights from Geary' test but made some extra innovations. The detailed comparison is not clear to me. I would expect the author to clarify this point in the related work section.
2. Some presentations can be improved (See Limitations section).

---

> ### Author Response · Authors · 2022-08-02
> **Response to Reviewer LJnR**
>
> Many thanks for the positive evaluation of the paper and for carefully checking our proofs. We now answer the raised questions.
>
>
> ### Comparison with Geary's test
>
> > The author pointed out this test share some insights from Geary' test but made some extra innovations. The detailed comparison is not clear to me. I would expect the author to clarify this point in the related work section.
>
> Geary’s test considers a univariate time series and counts the number of sign changes between consecutive observations; we pointed this out in the revised version of the paper. Our test differs from it for three main reasons:
>
> 1. we consider graphs data;
> 2. we allow for multivariate node signals;
> 3. we consider a weighted sum.
>
> Please note that in lines 87-90 we mention the improvements of what proposed w.r.t. to the Geary’s test.
>
>
>
> ### Statistical power for small graphs
>
> > The distribution of testing statistic is only characterized when the number of edges $|E|\to\infty$ Does it mean this test may not be powerful for small graph? Is there any possible idea to address this issue?
>
> Yes, if the number of observations (here, associated with the number of edges) decreases, then the power of the test is expected to decrease too. This is a characteristic of (virtually) every statistical test.
>
> That said, we mention that if we were to add arbitrary edges connecting nodes that are uncorrelated for the purpose of increasing $|E|$, then we would typically incur a decrease in the statistical power, as well.
>
> To keep the power of the test as high as possible, we suggest taking advantage of any prior information the user may have about the system in order to
>
> - discard all edges connecting nodes known to be independent
> - and keep only those edges that are most likely to link correlated nodes;
>
> for example, this may realize by removing edges that link sensors located too far apart in a sensor network.
>
> Please, see also Section D in the supplemental material.
>
>
>
> ### Use of the term "preprocessing"
>
> > In section 4, to perform whiteness test for dynamic and weighted graph, the author argued that we should pre-process it to make it static and then previous approach in Section 3 applies. Would this pre-processing procedure bring increase of type-I/type-II error for AZ-test? Is there any pre-processing procedure and which one is optimal? From my view point, this procedure seems too ad-hoc, and I hope to see some theoretical results behind this procedure.
>
> We apologize if we have misled the reviewer. With “preprocessing” (line 196) we simply mean that we created an _equivalent_ static representation of the dynamic graph, and by means of this equivalent static representation -- and the test in Eq. 2 -- we can perform a whiteness test also to dynamic graphs (which would have not been possible, otherwise). More specifically, line 198 refers to the argument of Section D in the supplemental material and our comment above.
>
>
> ### Other suggested improvements
>
> > Some notations and terminologies are introduced right after they first appeared, e.g., definition of $sng$, $E_\leftrightarrow$, $|E|$, $\to$. At least the author should create a notation subsection part in Section 1 to introduce all special notations that will be frequently used. If possible, the author may create a preliminary section in main content or supplementary to talk about basics of static/dynamic graph in detail, instead of briefly touching the basics in Section 3/4.
>
> > Assumptions (Ass1)-(Ass3) are important and repeatedly discussed in later section. The author are suggested to create a special "Assumption" environment to include them instead of putting them in Theorem 1. Also, comments of the assumption can be moved from supplementary to the main content if the paper gets accepted.
>
> We appreciate your valuable suggestions, thanks. We are now considering making such improvements, in compliance with the given space limitations which, at the time of submission, forced us to defer some discussions to the supplemental material.

---

### Official Review · Reviewer_Dqpj · 2022-07-11

**Rating:** 6
**Confidence:** 3
**Soundness:** 4 excellent
**Presentation:** 3 good
**Contribution:** 3 good

**Summary:**

The authors propose a novel model assessment tool by testing the whiteness (of the residuals) for dynamic graphs. Asymptotic normal null distribution (theorem 1) is developed, which enables easy calibration of the test.

**Questions:**

See above in weakness

**Limitations:**

No. Could we extend this test to graphs with negative weights? I am asking this since in practice the interactions between nodes could be both triggering effect or inhibiting effect.

**Strengths And Weaknesses:**

Strength:

The author proposed a statically solid testing method. Most importantly, the asymptotic null distribution is developed. Extensive numerical results help support the effectiveness of the method. Implementation is also provided to help recover the results.

Weakness:
1. (minor) notations: What is $\gamma$ in eq. (2)? I can see it is the testing threshold on line 122, but I think it should be clarified right after the equation; In theorem 1, I would prefer using A1 for Assumption 1 instead of Ass1.

2. What is the power of this test? I would expect the relation between power and (sample size, dimensionality). (PS: I do see the numerical assessment, but since you have the asymptotic null distribution, it would be better if you also have the (asymptotic) power. Or what is the technical difficulty is preventing you to give the power?)

3. (major) Although this method is novel in terms of the setting, I would expect numerical comparison to existing approaches for static graphs with the mentioned work in line 85.

---

> ### Author Response · Authors · 2022-08-02
> **Response to Reviewer Dqpj**
>
>
> Thanks for your positive feedback and suggested improvements.
>
>
> ### Comparison to existing approaches
>
> > (major) Although this method is novel in terms of the setting, I would expect numerical comparison to existing approaches for static graphs with the mentioned work in line 85.
>
> Unfortunately, we do not see any sound way for comparing our test with the work mentioned in line 85. The reason is that our whiteness test operates on a _single_ sample (one-sample test), while the referenced ones are all tests comparing the distribution of _two_ samples (two-sample tests). We are very happy to elaborate further on the concept if we have misunderstood the point you raised.
>
> ### Statistical power of the test
>
> > What is the power of this test? I would expect the relation between power and (sample size, dimensionality). (PS: I do see the numerical assessment, but since you have the asymptotic null distribution, it would be better if you also have the (asymptotic) power. Or what is the technical difficulty is preventing you to give the power?)
>
> We kindly refer the reviewer to Paragraph “Power of the test” in Section 5.1. There, you will find answers to your important questions.
>
>
> ### Notation
>
> > (minor) notations: What is $\gamma$ in eq. (2)? I can see it is the testing threshold on line 122, but I think it should be clarified right after the equation; In theorem 1, I would prefer using A1 for Assumption 1 instead of Ass1.
>
> Thanks, we intervened and addressed your points in the new revised paper.
>
>
> ### Graphs with negative weights
>
> > Could we extend this test to graphs with negative weights?
>
> Although we have not discussed this point in the paper, we may suggest two reasonable ways to operate with negative weights:
>
> - By applying the proposed test considering the absolute value of the weights;
> - By performing two tests: one considering only positive weights and setting to zero all negative ones, and the other setting to zero the positive weights and considering the absolute value of the negative ones. The results of the two tests can then be combined in different ways, e.g., via a Bonferroni correction.

---

### Official Review · Reviewer_z1xh · 2022-07-15

**Rating:** 6
**Confidence:** 3
**Soundness:** 2 fair
**Presentation:** 2 fair
**Contribution:** 2 fair

**Summary:**

The paper presents a whiteness hypothesis test for spatio-temporal graphs. The proposed AZ-test is an extension of traditional tests designed for system identification within graph signals, to detect dependencies among temporal observations as well as spatial dependencies among graph neighborhoods. Evaluation is performed on both synthetic as well as real datasets on problems of forecasting in spatio-temporal graphs by analyzing the prediction of residuals within the graph stream.

**Questions:**

1. The abstract and the first paragraph of section 4 mentions that the proposed work is applicable in the case of dynamic graphs ( allowing for changing connectivity patterns over time). This is handled by converting to a larger graph (Fig. 1) with nodes as a disjoint union of all the temporal nodes, (weighted) edges as a union of spatial and temporal edges. In the case that a new node v (and corresponding signal Xv[t]) is not present at time t, but added at at time point t+1, what is the corresponding value of Xv[t] used for forming the larger graph?

2. In the results in Table 2, the authors mention that "From the Temporal and Spatial variations of the AZ-test, we see that the temporal correlation appears more prominent than the spatial one for FCRNN". Is this based on the absolute values of C(G*) [without normalisation] in the two columns? If so, is this a fair comparison given that the number of spatial vs temporal edges can be vastly different in number?

**Limitations:**

There isn't a dedicated discussion on the limitations of the proposed framework. In this context, it would be good to consider and address the point mentioned in weaknesses about applicability and potential impact.

**Strengths And Weaknesses:**

STRENGTHS:

The paper is easy to follow and the presentation. The technical details and supporting theoretical analysis are clearly presented. The theoretical setup of extending the A-Z whiteness hypothesis test to (spatio-temporal) graphs is interesting and novel in formulation.

WEAKNESSES:

The applicability of the proposed framework beyond the considered forcasting setup is a bit unclear. If my understanding is correct, the null hypothesis test is constructed as a test for a zero mean and as well as spatially and temporally uncorrelated generating process. As such the test statistic C(G) does not distinguish between spatial and  temporal correlation patterns. Given that this is a rather strong assumption, I am unsure how such a test would be generally useful beyond examining residuals in the forecasting setting.

---

> ### Author Response · Authors · 2022-08-02
> **Response to Reviewer z1xh**
>
> We thank the reviewer for pointing out some parts of the manuscript worth improving.
>
> > it would be good to consider and address the point mentioned in weaknesses about applicability and potential impact.
>
> We hope to have properly addressed your points and that the following answers will shed light on your doubts.
>
>
> ### Applicability of the method beyond forecasting
>
> > The applicability of the proposed framework beyond the considered forcasting setup is a bit unclear.
>
> The proposed test is applicable beyond forecasting as it does not necessarily rely on temporal information. This fact is also reflected in the paper presentation, as we introduced the test first for static graphs in Section 3 and, then, for dynamic graphs in Section 4.
>
> ### Distinguishing between spatial and temporal correlation patterns
>
> > As such the test statistic C(G) does not distinguish between spatial and temporal correlation patterns. Given that this is a rather strong assumption, I am unsure how such a test would be generally useful beyond examining residuals in the forecasting setting.
>
> With our test statistic, we _can_ distinguish between spatial and temporal correlation patterns. Let us elaborate further.
>
> Please note that in the text (lines 191-194) we wrote:
>
> "_we show how to assign a weight_ $w_{tm}$ _to every temporal edge and yield balanced spatial and temporal contributions. In addition, Section C presents a straightforward way with which the user can decide how to trade off the impacts of the spatial and temporal edges_"
>
> and referred there to Section C of the supplemental material; as this was a derivation of the main text, details were moved there.
>
> Going back to your comment, the proposed test can take advantage of parameter $\lambda \in [0,1]$ (Eq. 14, supplemental material) to put more emphasis on the spatial correlation than the temporal one, or vice versa. For instance, in the experimental section (see, e.g., Table 2) we considered three “extreme” configurations for parameter $\lambda$:
> - $\lambda=1/2$. With this setup, spatial and temporal contributions (respectively, $\widetilde C_{sp}$ and $\widetilde C_{tm}$ in Eq. 14) are given the same relevance (identical weight).
> - $\lambda=1$. With this setup, we test the whiteness by inspecting the spatial contribution only.
> - $\lambda=0$ enables the test to inspect for the temporal correlation only.
>
> The user can select any $\lambda\in [0,1]$ he/she considers more appropriate as well as compute the test statistic for multiple $\lambda$s to better feel the correlation pattern at hand.
>
>
> ### Managing absent nodes
>
> > In the case that a new node v (and corresponding signal Xv[t]) is not present at time t, but added at at time point t+1, what is the corresponding value of Xv[t] used for forming the larger graph?
>
> If node $v$ appears at time $t+1$ but was not present at time $t$, then the node set $V^*$ of the “larger” graph $G^*$ would contain $v[t+1]$ but not $v[t]$ (ie, $v[t+1]\in V^*$, but $v[t]\not \in V^*$); accordingly, we don’t have – and we don’t need – a value for $\mathbf x_v[t]$.
> In other words, and looking at Figure 1, statistic $C(G^*)$ is built upon sum $\sum_{e\in E^*}$ over all well-defined edges $e$, that is, over every $e$ either of the form $e=(v[t], u[t])\in E_{sp}$ (thus, coming from edge set $E_t$ of Eq 6) or $e=(v[t],v[t+1])\in E_{tm}$, but only when $v$ is available at both time steps $t$ and $t+1$ (ie, $v[t],v[t+1]\in V^*$).
>
>
> ### Comparability of Spatio-temporal, Temporal, and Spatial statistics
>
> > Is this based on the absolute values of C(G*) [without normalisation] in the two columns? If so, is this a fair comparison given that the number of spatial vs temporal edges can be vastly different in number?
>
> Thanks for bringing up the point. All the three reported scores for the AZ-test denoted as “Spatio-temporal”, “Temporal”, and “Spatial” are comparable to each other. In fact, please note that the three scores are computed as $C(G^*;1/2), C(G^*;0),$ and $C(G^*;1)$, respectively, from the same function $C(G^*;\lambda)$ defined in Eq 14 of the supplemental material (we kept details in there). It then follows from Theorem 1 that all the three scores are (asymptotically) distributed as a standard Gaussian distribution $N(0,1)$ (see also our comment in lines 527-529 of the supplemental material). We improved the text of Section 5.2.2 in the revised version of the manuscript to improve clarity.

---

> > ### Comment · Reviewer_z1xh · 2022-08-06
> > **Response to Author Reply**
> >
> > Thank you for your efforts in the rebuttal and for the clarifications provided. Since the authors have addressed my main concerns in the review, I am willing to increase my score to weak accept based on the response.
> >
> > A small suggestion would be to include the explanations provided for the questions above either in the main paper or supplementary, especially the point on the comparability of the spatial, temporal and spatio-temporal of C(G*).

---

> > > ### Author Response · Authors · 2022-08-08
> > > **Explainations added**
> > >
> > > We followed your suggestion and added the above clarifications in the revised version of the paper. We are happy to further improve the paper presentation if there is anything else you would like us to add in.
> > >
> > > We appreciate your willingness to increase the score of our paper, and hope you will update it at the due time.

---

### Meta-Review · Area_Chair_dVYi · 2022-08-27

**Recommendation:** Accept
**Confidence:** Certain

**Metareview:**

The paper studies the whitness hypothesis test for spatio-temporal graph, which is a fundamental problem and can be relevant to many machine learning tasks. The authors have done a great job in the rebuttal phase in addressing reviewers’ comments. I believe it is a worthwhile paper to be published in NeurIPS.

**Award:**

No

---

### Decision · Program_Chairs · 2022-09-14

Accept